# Association of Public Sports Space Perception with Health-Related Quality of Life in Middle-Aged and Older Adults—Evidence from a Survey in Shandong, China

**DOI:** 10.3390/bs13090736

**Published:** 2023-09-04

**Authors:** Chenchen Liu, Yan Gao, Zhihao Jia, Liangyu Zhao

**Affiliations:** 1School of Medical Information Engineering, Jining Medical University, Jining 272067, China; lchenchen1979@hotmail.com; 2School of Physical Education, Shandong University, Jinan 250061, China; jiazhihao@mail.sdu.edu.cn (Z.J.); zhaoliangyu@mail.sdu.edu.cn (L.Z.)

**Keywords:** public sports space perception, middle-aged and older adults, health-related quality of life, physical exercise activities

## Abstract

Creating a healthy living environment for middle-aged and older adults is a key strategy for countries to address the aging challenge, but the effects of such an environment on the health-related quality of life (HRQoL) of middle-aged and older adults remain underexplored. This study aimed to examine the link between public sports facilities and the HRQoL of middle-aged and older adult residents in communities. A total of 1169 respondents (average age: 66.84; male: 46.19%) were selected from the Shandong, China. This study measured respondents ‘physical activity (PA) using the International Physical Activity Questionnaire, the HRQoL of respondents using the 36-item Short Form Health Survey, and the Public Sports Space Perception Scale for respondents’ public sports space perception. Correlation analysis and logistic regression analysis were employed to test the relationship among public sports space perception, physical activity (PA), and HRQoL. The mediating role of PA was conducted using the PROCESS macro for SPSS. The results revealed that public sports space perception only influenced the HRQoL of middle-aged and older adults through light-intensity PA (PCS: B  =  0.09, 95% CI 0.01, 0.03; MCS: B  =  0.02, 95% CI 0.01, 0.05) among light-intensity PA, moderate-intensity PA, vigorous-intensity PA., and this mediation model varied across different age groups of middle-aged and older adults. Moreover, digital inclusion only moderated the psychological aspect of HRQoL of middle-aged and older adults (*p* < 0.05). This study provided empirical evidence for enhancing the HRQoL of middle-aged and older adults and offered useful insights for the planning and design of public sports facilities and the formulation of health management policies for middle-aged and older adults.

## 1. Introduction

Population aging is a global phenomenon of demographic transition, and China has officially adopted a national strategy to actively cope with it. Active and healthy aging not only aims to prolong the biological age of middle-aged and older adults but also to enhance their quality of life and life satisfaction [1]. Health-related quality of life (HRQoL) is a comprehensive assessment of the health status of middle-aged and older adults across multiple domains and levels, which is directly linked to health outcomes and reflects a person’s health level better than life expectancy and mortality [2]. Identifying the factors associated with HRQoL is vital for improving the health and well-being of middle-aged and older adults [3,4]. Therefore, delving into the factors that influence the HRQoL of middle-aged and older adults has far-reaching implications for promoting healthy aging and active aging and can offer theoretical guidance for countries to devise relevant policies.

Physical activity (PA) of middle-aged and older adults is a key factor that influences their HRQoL, and its relationship with HRQoL has been established [4]. PA included three intensities, namely light-intensity PA (LPA), moderate-intensity PA (MPA), and vigorous-intensity PA (VPA). PA offers various benefits for middle-aged and older adults, such as reducing the risk of chronic diseases and premature mortality [5]. To achieve substantial health benefits, the US PA guidelines recommend at least 150 min of moderate or 75 min of vigorous aerobic exercise per week or a combination of both [6]. Previous studies have shown that physical activities, such as physical exercise, affect the physical and mental health of middle-aged and older adult residents in communities and can predict their health changes [7]. Furthermore, physical exercise is a modifiable factor that has been frequently used as an intervention and a mediator for the health of middle-aged and older adults in prior studies. It has significant public health and clinical implications for enhancing the quality of life and fostering active aging of middle-aged and older adults. Therefore, the first hypothesis of this study is that physical activity will have a positive impact on health-related quality of life in middle-aged and older adults.

In the era of actively coping with population aging, how to speed up the development of health-promoting environments has become a valuable research topic. Countries have adopted the creation of suitable environments for middle-aged and older adults and healthy living environments as one of the main strategies to tackle the aging issue [8]. Existing studies have explored the relationship between health-supportive environments, such as physical activity-built environments, healthy communities, residents’ health, and physical activity. Studies have shown that the successful aging of middle-aged and older adults and the improved quality of life requires the support of the objective environment [9,10,11] and its correlation with factors such as obesity, BMI, blood pressure, weight status, mental health, and overall well-being in the middle-aged and older adults has been confirmed by [10,11,12]. In addition, some indicators in the built environment, such as the number of sports facilities, the site area, and the distance of leisure places, are closely related to residents’ health behavior and health [11,13]. However, the research on the relationship between public sports spaces and residents’ health has mostly used built environment indicators (neighborhood environment) as proxies and has not examined sports spaces as independent entities [14]. Although previous studies on built environments (including sports space construction indicators) have found that the perception of built environments affects residents’ exercise behavior and health [13] and that its impact may be larger than that of objective built environments on residents’ health [15], this may not fully capture the impact pathways and mechanisms of public sports spaces on residents’ health. On the other hand, in practical terms, prior studies have paid more attention to objective physical space construction and have largely overlooked residents’ subjective perceptions and actual needs of public spaces. The traditional public sports facilities planning practice has emphasized supply over demand orientation, resulting in a mismatch between supply and demand [16]. Therefore, this study proposes a second hypothesis: that public sports space perception will promote the participation of middle-aged and older adults in physical exercise activities and improve the health-related quality of life of middle-aged and older adults.

Physical exercise activity starts from physical exercise, and although it is closely related to sports space, the association with health will be affected differently by age. Studies suggest that this association may decrease [17,18] with age due to increased physical exercise activity with increased leisure time after retirement with older adults. Therefore, it is reasonable to propose the third hypothesis of this study that the mediating role of physical exercise activity in public sports spaces and health-related quality of life in community older adults may show different outcomes in older adults of different ages.

In addition, changes such as COVID-19 have become an incentive to encourage middle-aged and elderly people to use digital technology and narrow the digital divide [19]. Compared with traditional media, digital integration will significantly affect the health of middle-aged and older adults [20], play an important role in promoting health, social participation, and security, and is also an important way to improve the quality of life of middle-aged and older adults, increase happiness, and promote active aging [21,22,23]. Digital integration can also improve the perception of public sports space [24]. Digital technology can provide more real-time information, such as air quality, weather conditions, and public facilities, which can help middle-aged and older adults to better understand the use and safety of public sports spaces and improve their perception of public sports spaces. Therefore, we propose the fourth hypothesis of this study, that digital integration has a modulatory role in the impact of the perception of the public sports space on health-related quality of life for seniors in the community.

As China faces an accelerating and deepening aging challenge, the health issues arising from various built environments have not been adequately addressed and resolved, leading to the gradual accumulation of problems such as public health and fitness behavior changes, which cannot be solved in a short span of time. China needs to expedite the research on the relationship between the built environment and health and elucidate the specific pathways and mechanisms of how public sports spaces affect the health of middle-aged and older adults based on sustainable development. Moreover, in the context of the post-pandemic era, the extensive use of smart devices in various public environments and places also reveals the social problem of middle-aged and older adults’ “disconnection” from the digital age [25]. The investigation of the impact of public sports spaces on the health of middle-aged and older adults cannot ignore this objective temporal background. Therefore, this study takes China’s population aging and digitalization in the post-pandemic era as the research backdrop and focuses on the relationship between sports spaces and HRQoL of middle-aged and older adults as the research subject. The theoretical model of this study was constructed based on the above discussion (Figure 1), and it delves into the impact mechanism of public sports space subjective perception on the HRQoL of community middle-aged and older adults, aiming to clarify the pathways of how public sports space perception influences the HRQoL of community middle-aged and older adults.

## 2. Materials and Methods

### 2.1. Participants

This study involved 2000 middle-aged and older adults who lived in 48 communities across four cities in Shandong Province (Jinan, Qingdao, Yantai, and Tai’an). All respondents were older than 45 years. These cities had a high proportion of middle-aged and older adults population, with more than 14% of their residents aged 65 or above. We collected data from March to June 2023 using a questionnaire that consisted of five sections: demographic information, perception of public sports spaces, physical activity behavior, HRQoL, and digital inclusion level.

We distributed 1000 paper questionnaires and 1000 electronic questionnaires and received 902 and 943 responses, respectively, resulting in a response rate of 92.25%. We screened the valid questionnaires based on the criteria of data completeness, answer consistency, and adherence to the answering rules and obtained 1669 valid questionnaires, with an effective response rate of 90.55%. We ensured the anonymity and confidentiality of the participants and the data throughout the data collection process. We excluded the questionnaires with missing data, biased answers, or irregular answering patterns and obtained 1406 eligible participants. However, due to data format issues, we lost 376 original data records and only retained 1030 complete data records of community-dwelling middle-aged and older adults in Shandong Province. We further excluded the participants who were out of the age range, had physical disabilities, memory impairments, or mental illnesses, and ended up with 869 qualified data records. Therefore, this study included the complete data of 869 participants. Each participant gave written informed consent before joining the study. Written informed consent was indicated on the questionnaire, and the middle-aged and older adults participated entirely voluntarily and anonymously.

### 2.2. Public Sports Space Perception

We used the Public Sports Space Perception Scale to measure how residents perceived public sports spaces. This scale was developed by Cai Yujun et al. from the Shanghai University of Sport and consisted of 20 items across five dimensions: reachability, accessibility, stoppability, service level, and emotional stimulation. The scale employed a Likert 5-point rating method, where 1–5 indicated “very inconsistent” to “very consistent.” The total score was the sum of all items, and a higher score indicated a better perception of public sports spaces. Previous studies have demonstrated that this scale had a high reliability with a Cronbach’s α coefficient of 0.878 [16].

### 2.3. Physical Activity

The International Physical Activity Questionnaire (IPAQ) mainly records physical activities related to work, transportation, household chores, recreation, or exercise. It asks the respondents about their specific activities in the previous week, the total duration and number of days for each activity, and whether the activity was vigorous, moderate, or light. The questionnaire was developed by the World Health Organization and other international organizations, using an objective and standardized assessment method. It comprehensively evaluates various physical activities and classifies the physical activity level according to the assessment results. The IPAQ questionnaire has the advantages of being simple to operate, easy to implement, and widely applicable. It has been widely used in the assessment and research of physical activity levels worldwide. In addition, the questionnaire has been used for research in the Chinese population and has undergone reliability and validity tests in the Chinese middle-aged and older adults population, showing acceptable reliability and applicability [26].

### 2.4. HRQoL

HRQoL is the perceived physical and mental health of an individual or group over time, including both physical component summary (PCS) and mental component summary(MCS). The HRQoL scale used the 36-item Short Form Health Survey. The 36-item Short Form Health Survey (SF-36) is a commonly used medical health survey tool for assessing an individual’s physical health and quality of life. The survey contains 36 questions covering various aspects of physiological, psychological, and social health, such as bodily pain, vitality, physical functioning, and social roles. Physical function, body roles, body pain, and general perceptions of health were calculated as PCS, and mental health, vitality, emotional role, and social functioning were calculated as MCS. The SF-36 has been widely used worldwide and has become one of the standard tools for evaluating an individual’s health and quality of life. The survey has many advantages, such as being simple to operate, easy to implement, stable in measurement results, high in validity, etc. It can be used for health surveys and comparative studies of different populations, cultural backgrounds, and language groups. In addition, the questionnaire has been used for research in the Chinese population and has undergone reliability and validity tests, showing acceptable reliability and applicability. Therefore, it can be used as a reliable tool for studying and evaluating the HRQoL of middle-aged and older adults [27].

### 2.5. Digital Inclusion

The survey of middle-aged and older adults’ digital inclusion level was based on the China Comprehensive Social Survey and combined with previous relevant studies, measured by three dimensions: digital access, digital skills, and digital output. Digital access considered whether there was internet access at home and whether there were mobile phones, tablets, and computers; digital skills considered six aspects: opening websites, downloading and installing apps, searching for information, identifying information authenticity, expressing ideas, and online payment; digital output considered six aspects: social activities, self-presentation, online action, leisure and entertainment, information acquisition and business transactions, and finally calculated the mean value as the score. The validity of this way of surveying middle-aged and older adults’ digital inclusion level in China has been confirmed, and it has good reliability and validity [28,29].

### 2.6. Socio-Demographic Variables

Based on previous studies, the covariates in this study included age, gender (male or female), residence (rural or urban community), household registration (non-agricultural or rural), marital status (married or single; divorced or widowed), education level (“illiterate (never attended school),” below primary school, junior high school, high school, vocational school and college or above), family annual income, and health status. Health status included smoking status (non-smoker and smoker), drinking status (non-drinker and drinker), and chronic disease status (“never had chronic disease,” “only had one chronic disease,” and “had two or more chronic diseases”).

### 2.7. Statistical Analyses

First, we used SPSS 23.0 (IBM Corp., Armonk, NY, USA) statistical software to perform descriptive statistical analysis on the collected valid questionnaires. We tested the age differences of public sports space perception, physical exercise activity, digital inclusion, and HRQoL by analysis of variance and analyzed and compared the different demographic characteristics and differences in each variable. Then, in order to test the mediating role of physical exercise activity on public sports space and HRQoL, we first used the Pearson product-moment correlation coefficient to preliminarily verify the hypotheses, which process was performed using Origin 2023 (Origin Lab. Northampton, MA, USA).

To ensure the accuracy of the results and to address the limitation of cross-sectional analysis that may lead to bias in assessing mediation, we ran alternative models to test the hypotheses related to public sports space, physical exercise activity, and HRQoL. All covariates were entered into the logistic regression model. Alternative model 1 verified the association between public sports space and physical exercise activity, alternative model 2 verified the correlation between physical exercise activity and HRQoL and depression, and alternative model 3 verified the correlation between public sports space and HRQoL and depression.

Finally, we tested and bootstrap analyzed the mediating role of physical exercise activity variable between public sports space perception and community-dwelling middle-aged and older adults’ HRQoL using the PROCESS plug-in of SPSS 23.0, with 5000 Bootstrap samples. Finally, we analyzed the moderating role of digital inclusion on the relationship between public sports space perception and community-dwelling older adults’ HRQoL using the model I in Hayes’ PROCESS macro in SPSS 23.0.

## 3. Results

### 3.1. Descriptive Analysis

Appendix A shows the sociodemographic characteristics of the participants (as shown in Appendix A). Our study included 1169 middle-aged and older adults (mean age, 56.84 years; standard deviation [SD] 10.23), including 540 males (approximately 46.19%) and 629 women (approximately 53.81%). Currently, the proportions of community-dwelling middle-aged and older adults with junior college, bachelor’s, master’s, and doctoral degrees were 8.20%, 14.55%, 1.04%, and 0.69%, respectively. Among the community-dwelling middle-aged and older adults, more than 50% of them had retired, 30% were unemployed (including those without jobs), and nearly 20% were still working. Smokers accounted for the largest proportion of community-dwelling middle-aged and older adults, at 72.29%, while quitters and never-smokers accounted for smaller proportions, at 11.20% and 16.51%, respectively. For drinking status, more than half of the community-dwelling middle-aged and older adults (53.93%) did not drink, and among those who drank, the largest proportion (30.48%) drank more than once a month. In addition, 52.89% of the community-dwelling middle-aged and older adults had at least one chronic disease, with the highest proportion having one chronic disease, at 20.55%, and the proportions having two, three, and four or more chronic diseases were 12.24%, 8.31%, and 4.62%, respectively. The mean score of public sports space perception for community-dwelling middle-aged and older adults (including the young-old population) in Shandong Province was 3.54, which exceeded the median value.

### 3.2. Correlation Analysis

We performed Pearson correlation analysis on community-dwelling middle-aged and older adults’ public sports space perception and physical exercise activity. As shown in Appendix A, public sports space perception was significantly correlated with physical exercise activity participation time and frequency and not correlated with various physical exercise activity amounts of community-dwelling middle-aged and older adults. Community-dwelling older adults’ public sports space perception and its dimensions of accessibility perception, inclusiveness perception, stoppability perception, service level perception, and emotional stimulation perception were significantly correlated with LPA, MPA, and VPA participation time and frequency (*p* < 0.001). This indicates that the stronger the community-dwelling middle-aged and older adults ’ perception of public sports space, the more frequently and longer they participate in LPA, VPA, and MPA.

We performed Pearson correlation analysis on community-dwelling middle-aged and older adults’ public sports space perception and HRQoL and its dimension scores. As shown in Appendix A, public sports space perception was significantly correlated with the HRQoL and its dimension scores (*p* < 0.001). Community-dwelling middle-aged and older adults’ public sports space perception had a significant positive correlation with the HRQoL and its dimension scores (*p* < 0.001), indicating that the stronger the community-dwelling middle-aged and older adults’ perception of public sports space, the better their HRQoL.

We performed Pearson correlation analysis on community-dwelling middle-aged and older adults’ physical exercise activity and HRQoL and its dimension scores. As shown in Appendix A, the correlation between physical exercise activity and HRQoL and its dimension scores was not consistent. Only LPA participation frequency and average physical exercise participation frequency were significantly correlated with all dimensions of community-dwelling middle-aged and older adults’ HRQoL (*p* < 0.001), indicating that the more frequent the community-dwelling older adults’ LPA participation, the better their HRQoL. Specifically, LPA frequency had a positive correlation with 16 items: LPA time, VPA frequency, VPA time, MPA frequency, MPA time, average exercise frequency, physical functioning, role-physical, bodily pain, general health, vitality, social functioning, role-emotional, mental health, physical component score, and mental component score. The impact of activity amount on HRQoL was lower than that of participation frequency and time on HRQoL.

In the alternative models, all covariates were entered into alternative models 1–3, and the study controlled for gender, age group, education, retirement, marriage, economic condition, drinking, smoking, chronic disease, and other corresponding variables. As shown in Appendix A, alternative model 1 verified the association between public sports space perception and its dimensions and physical exercise activity. It indicated that community-dwelling middle-aged and older adults’ public sports space perception could significantly positively affect physical exercise activity (B = 0.59, *p* < 0.01) and had a significantly greater impact on LPA than other intensity physical exercise activities. Alternative model 2 verified the association between public sports space perception and its dimensions and HRQoL and its dimensions. The results showed that community-dwelling middle-aged and older adults ’ public sports space perception could significantly positively affect HRQoL (B = 0.16, *p* < 0.01). Alternative model 3 verified the association between community-dwelling middle-aged and older adults ’ physical exercise activity and HRQoL and its dimensions. The results showed that community-dwelling middle-aged and older adults ’ physical exercise activity could significantly positively affect the HRQoL, but this physical exercise activity was only LPA (B = 0.16, *p* < 0.01). LPA and VPA did not have a significant impact (B = 0.16, *p* > 0.01).

In summary, the study found that only the variables of sports space perception, light physical exercise frequency, and physical component score and mental component score of HRQoL had a significant correlation, meeting the requirements for being a mediator variable, so it could be considered whether there was a mediation effect. The rest of the physical exercise activity variables did not meet the prerequisite conditions for the mediation test and were not included in the consideration of this study in the follow-up research.

### 3.3. Mediator Model Test

This article uses the PROCESS plug-in Model 4 (Model 4 is a simple mediation model) developed by Hayes (2012) to analyze the mediation effect of LPA on the relationship between community middle-aged and older adults’ perception of public sports space and health-related quality of life, controlling for gender, age, education, retirement, marriage, economic condition, drinking, smoking, and chronic diseases. The results are shown in Table 1 and Table 2.

According to Table 1, the results show that the perception of public sports space by community middle-aged and older adults has a significant positive predictive effect on PCS (β = 0.054, *p* < 0.01). After introducing the mediator variable, the positive predictive effect of public sports space perception on PCS remains significant (β = 0.052, *p* < 0.01). The perception of public sports space has a significant positive predictive effect on the frequency of LPA (β = 0.024, *p* < 0.01), and the frequency of LPA has a significant positive predictive effect on PCS (β = 0.266, *p* < 0.01).

According to Table 2, the results show that the perception of public sports space by community middle-aged and older adults has a significant positive predictive effect on MCS (β = 0.193, *p* < 0.01). After introducing the mediator variable, the positive predictive effect of public sports space perception on MCS remains significant (β = 0.182, *p* < 0.01). The perception of public sports space has a significant positive predictive effect on the frequency of LPA (β = 0.026, *p* < 0.01), and the frequency of LPA has a significant positive predictive effect on MCS (β = 0.850, *p* < 0.01).

In summary, LPA is an important mediator factor for the influence of public sports space perception on health-related quality of life. The simple mediation model is shown in Figure 2.

### 3.4. Modulation Model Test

This study also verifies the moderating effect of digital integration on the relationship between public sports space perception and health-related quality of life. From Table 3, it can be seen that testing the moderating effect of digital integration on the relationship between public sports space perception and PCS requires three models (Model 1, Model 2, and Model 3). Model 1 includes public sports space perception, as well as control variables such as gender, age group, education, retirement, marriage, economic condition, drinking, smoking, and chronic diseases. Model 2 adds digital integration to Model 1, and Model 3 adds an interaction term (the product of digital integration and public sports space perception) to Model 2. The purpose of Model 1 is to study the effect of public sports space perception on PCS without considering the interference of digital integration. Public sports space perception and PCS show significance (B = 0.103, *p* = 0.000 < 0.01). This means that public sports space perception has a significant impact on PCS.

The moderating effect can be known by looking at the significance of the interaction term in Model 3. From Table 4, it can be seen that the interaction term of public sports space perception and digital integration does not show significance (B = 0.002, *p* > 0.05), and from Model 1, it can be seen that public sports space perception has an impact on PCS. This means that when public sports space perception affects PCS, the impact magnitude of digital integration at different levels remains consistent.

Testing the moderating effect of digital integration on the relationship between public sports space perception and MCS requires three models (Model 4, Model 5, and Model 6). Model 4 includes public sports space perception, as well as control variables such as gender, age group, education, retirement, marriage, economic condition, drinking, smoking, and chronic diseases. Model 5 adds digital integration to Model 4, and Model 6 adds an interaction term (the product of public sports space perception and digital integration) to Model 5. The purpose of Model 4 is to study the effect of public sports space perception on MCS without considering the interference of digital integration. From Table 4, it can be seen that public sports space perception shows significance (B = 0.151, *p* = 0.000 < 0.01). This means that public sports space perception has a significant impact on MCS. The interaction term of public sports space perception and digital integration shows significance (B = 0.009, *p* = 0.045 < 0.05). This means that when public sports space perception affects MCS, the different levels of digital integration have significant differences in the impact magnitude.

In summary, digital integration is an important moderating variable that affects the MCS of HRQoL in the relationship between public sports space perception and health-related quality of life, as shown in Figure 3.

### 3.5. Analysis of Heterogeneity

By studying the differences in the effects of public sports space perception and physical exercise activities on the HRQoL of middle-aged and older adults in different age stages in China (variance analysis see Appendix A), the results show that there are obvious differences in the impact paths of public sports space perception and physical exercise activities on the HRQoL among middle-aged and older adults in different age stages in China (see Table 4).

For public sports space perception and LPA, no matter what age stage the middle-aged and older adults are in, walking is always affected by public sports space perception, but for people aged 60–69, although LPA is also affected by sports space perception, the significance is only significant at the 0.05 level. For LPA and PCS, LPA has a significant positive impact on the PCS of middle-aged and older adults aged 60 and above (60–69: β = 0.635, *p* < 0.01; 70–80: β = 0.623, *p* < 0.05), but LPA has no significant impact on the PCS of middle-aged and older adults aged 40–59 (40–59: β = 0.307, *p* > 0.05). For public sports space perception and PCS, public sports space perception has a significant positive impact on the PCS of middle-aged and older adults in any age stage (50–49: β = 0.063, *p* < 0.01; 60–69: β = 0.122, *p* < 0.01; 70–80: β = 0.149, *p* < 0.05).

For LPA and MCS, LPA has a significant positive impact on the MCS of middle-aged and older adults aged 60 and above (60–69: β = 1.180, *p* < 0.01; 70–80: β = 1.232, *p* < 0.05), but LPA has no significant impact on the MCS of middle-aged and older adults aged 40–59 (β = 0.442, *p* > 0.05). For public sports space perception and MCS, public sports space perception has no significant impact on the MCS of middle-aged and older adults aged 60–69 (β = 0.065, *p* < 0.01), but public sports space perception has a significant impact on the MCS of middle-aged and older adults aged 70–80 (β = 0.305, *p* < 0.05).

The mediation effect also has significant differences among middle-aged and older adults in different age stages in China. For middle-aged and older adults aged 40–59, the mediation path of public sports space perception affecting HRQoL through physical exercise activities does not exist. For middle-aged and older adults aged 60–69, public sports space perception can affect HRQoL through physical exercise activities. LPA plays a partial mediating role in the path of public sports space perception affecting PCS and a complete mediating role in the path of public sports space perception affecting MCS. For middle-aged and older adults aged 70–80, public sports space perception can affect the HRQoL through physical exercise activities. LPA plays a partial mediating role in the path of public sports space perception affecting HRQoL.

## 4. Discussion

This study explores the impact of public sports space perception on the HRQoL of community middle-aged and older adults and analyzes the mechanism of how public sports space perception promotes the HRQoL of community middle-aged and older adults. LPA has a significant mediating effect between public sports space perception and the HRQoL of middle-aged and older adults, which is similar to the conclusions of previous studies [30]. Although this study only found the mediating role of LPA, the results of this study show that public sports space perception can improve the HRQoL of middle-aged and older adults through physical exercise activities, and physical exercise activities are an effective way for public sports space perception to affect the HRQoL of middle-aged and older adults.

The mean values of PCS and MCS of middle-aged and older adults are low and do not change much. The mean values (standard deviations) of the PCS and MCS are 51.691 (8.308) and 43.01 (12.994), respectively. PCS scores were lower with increasing age, while MCS developed in waves, similar to a previous study with a Chinese population [31].In addition, our study is similar to the findings of studies with older adults. The mean values (standard deviations) of the PCS and MCS of urban solitary middle-aged and older adults in Shaanxi Province are 46.01 (6.69) and 42.05 (11.5), respectively [32]. Generally speaking, due to the large differences in living environments among middle-aged and older adults in different provinces in China and the different focuses of government policies for middle-aged and older adults, there is no consistent conclusion on whether the physical health condition of middle-aged and older adults is better than their psychological health, or vice versa. Therefore, it is necessary to compare and analyze the HRQoL of middle-aged and older adults in different provinces in China in the future and to explore the health inequality problem caused by regional differences.

Although currently, there is not enough direct evidence on the relationship between public sports space perception and HRQoL, existing studies have explored the relationship between the perceived built environment and people’s health. A study of community residents aged 60 and above in Brazil showed that the perceived environment index was positively correlated with health [33]. An empirical study of middle-aged and older adult women found that subjective perceived built environment was negatively correlated with obesity among middle-aged and older adult women [34]. In addition, a cross-national study of middle-aged and older adults found that participating in moderate and vigorous physical activities was associated with better physical function, better mental health, and better quality of life [35]. Therefore, theoretically, the mediating role of physical exercise activities between public sports space perception and the HRQoL of middle-aged and older adults exists. However, the size of this mediation effect may vary depending on factors such as physical activity type and individual characteristics.

Similar to other studies, related studies have shown that the key mediator between public sports space perception and health may be physical activity [36]. For example, a study of middle-aged and older adults in Hong Kong, China, found that public sports space perception was positively correlated with increased physical activity level, and increased physical activity level could improve the quality of life of middle-aged and older adults [37]. Compared with previous studies, this study has the following characteristics. This study analyzed the relationship between different types of physical activities and public sports space perception and HRQoL, considering individual characteristic factors, and found that only LPA met the premise assumption of the mediation model.

Affected by the physical condition and needs of middle-aged and middle-aged and older adults, the mediating role of LPA varies in different age stages. Specifically, in the early old age group, LPA did not play a mediating role. For the path of sports space perception affecting PCS through LPA, the older the age, the greater the mediation effect of LPA. For the path of sports space perception affecting MCS through LPA, LPA played a complete mediating role in the middle-aged and older adults group aged 60–69 and a partial mediating role in the middle-aged and older adults group aged 70–80. Similar to previous studies, previous studies believed that the reason for the age difference was related to the living habits and physiological and psychological differences of middle-aged and older adults in different age stages. First, the physical exercise of retired people has an increasing trend, which may be one of the reasons why this path is not significant in the early old age group [38,39]. Second, with the increase of age, the physiological ability and function of middle-aged and older adults gradually decline, such as the decrease in muscle mass and strength and illness. These physiological changes may lead to people readjusting their exercise intensity and frequency during exercise, thus causing differences in the impact path on the HRQoL [40]. In addition, from a psychological perspective, different age stages have different perspectives on public sports space perception, which may also affect the path of public sports space perception, affecting the HRQoL of middle-aged and older adults. Studies have shown that younger middle-aged and older adults pay more attention to the social and interactive aspects of public space [41,42].

In addition, this study also explored the moderating role of digital integration on the relationship between public sports space perception and HRQoL of community middle-aged and older adults. The results showed that digital integration did not play a significant moderating role in the physical health aspect of the HRQoL of community middle-aged and older adults’ public sports space perception. This conclusion is somewhat different from previous studies. One possible reason is that digital technology has been more deeply applied in China after experiencing COVID-19, which may be one of the reasons for the inconsistency between our results and other studies [43]. Previous studies have shown that the use of digital technology has an important promoting effect on improving people’s health, social participation, and physical exercise [20,21,22,23,44]. In terms of physical health, reasonable use of the internet and browsing online health information have a positive impact on enhancing health awareness and increasing health knowledge, which is conducive to improving the health literacy of middle-aged and older adults. Although the results of this study did not support the role of digital integration in the physical health of middle-aged and older adults, this study found that positive public sports space perception helps to improve the HRQoL of community middle-aged and older adults, and digital integration factors played a certain moderating role in it. Compared with middle-aged and older adults with low digital integration, middle-aged and older adults with high digital integration have better mental health [45]. Digital integration can significantly improve the performance of the middle-aged and older adults group in terms of mental health, help to improve their mood, alleviate negative psychological feelings such as loneliness and isolation, and enhance their social adaptability and subjective well-being [22]. Therefore, for community middle-aged and older adults with high digital integration, the improvement of public sports space perception has a greater positive impact on HRQoL.

The study still has some shortcomings and limitations. On the one hand, there is the limitation of causality. Although the study found that public sports space perception had a significant positive impact on HRQoL, the specific causal relationship between them is still unclear. Future studies can explore the causal relationship between public sports space perception and HRQoL through longitudinal research design. On the other hand, there is the limitation of influencing factors. There are many factors that affect public sports space perception, behavior, and health. However, in this study, due to the limitations of research conditions and focus, not all relevant factors were considered as control variables or moderating variables, which may lead to some errors in the research results. This may be one of the reasons why the impact coefficient of the mediation effect in this study is small. Future studies should further explore the relationship between social environment, natural environment, built environment, and middle-aged and older adults’ health behavior and health on the basis of controlling these factors. Therefore, in the future, these factors should be deeply analyzed, and the theoretical model should be continuously expanded and improved. Finally, this study used the World Health Organization’s age division of middle-aged and older adults, but it was limited by China’s national conditions when subdividing consecutive age groups (China has a retirement system of 60 years). Therefore, the results of this study cannot be generalized to other ethnic groups.

## 5. Conclusions

This study constructs a model of how public sports space perception affects the HRQoL of community middle-aged and older adults and, through empirical analysis, obtains the following conclusions: the perception of public sports space can significantly affect the HRQoL of community middle-aged and older adults. Among the different intensities of physical exercise activities, only LPA plays a partial mediating role in the process of public sports space perception affecting the HRQoL of community middle-aged and older adults. The mediating role of LPA varies among different age groups; digital integration only plays a moderating role in the process of public sports space perception, affecting the mental health of community middle-aged and older adults.

## Figures and Tables

**Figure 1 behavsci-13-00736-f001:**
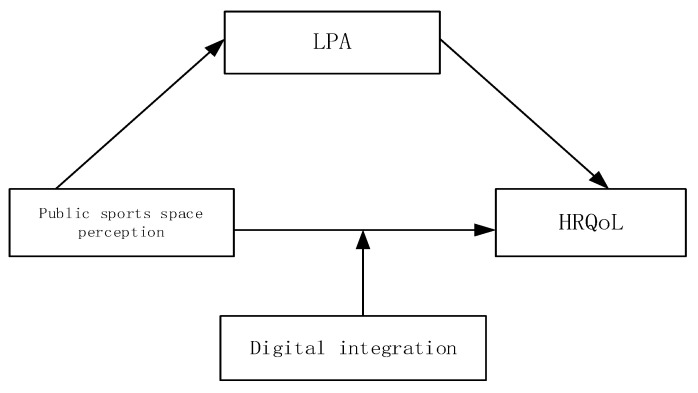
Research design.

**Figure 2 behavsci-13-00736-f002:**
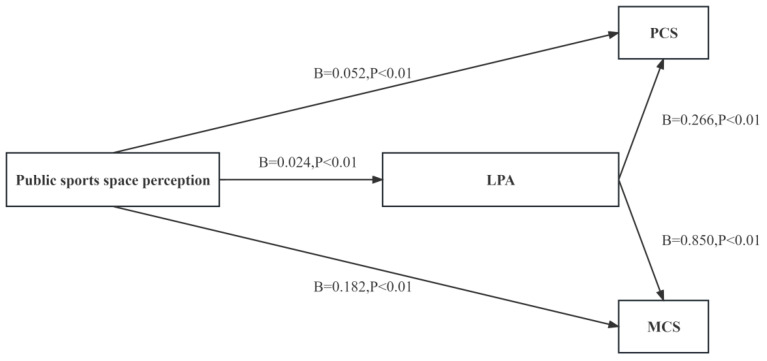
A mediation model of the association between public sports space perception and HRQoL through LPA. Path coefficients are shown. Note: Mediation analyses were performed with 5000 bias-correct bootstrapped samples.

**Figure 3 behavsci-13-00736-f003:**
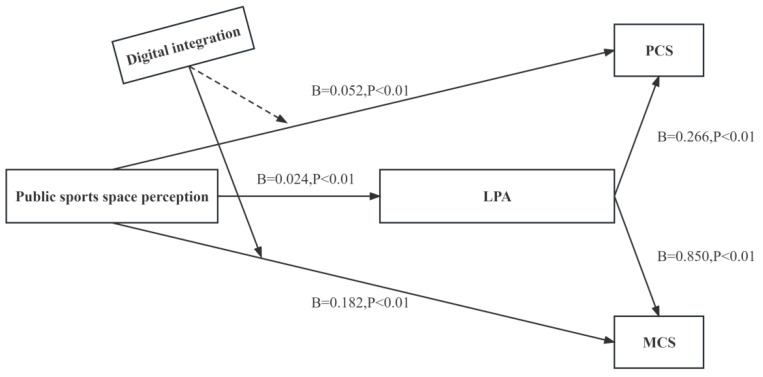
A modulation model. Path coefficients are shown.

**Table 1 behavsci-13-00736-t001:** The mediation role of mild physical exercise activity between the perception of public sports space and body composition scoring.

Variables	PCS	LPA	PCS
Public sports space perception	0.054 **	0.024 **	0.052 **
LPA	—	—	0.266 **
R^2^	0.151	0.102	0.250
Adjust R^2^	0.124	0.074	0.228
F	F (12, 381) = 5.657, *p* = 0.000	F (12, 854) = 2.015, *p* = 0.000	F (13, 380) = 5.253, *p* = 0.000

** *p* < 0.01, —indicates that there was no linear regression between the two variables.

**Table 2 behavsci-13-00736-t002:** The mediation role of mild physical exercise activity between the perception of public sports space and MCS.

Variable	MCS	LPA	MCS
Public sports space perception	0.193 **	0.024 **	0.182 **
LPA	—	—	0.850 **
R^2^	0.126	0.041	0.133
Adjust R^2^	0.110	0.024	0.115
F	F (7, 368) = 7.958, *p* = 0.000	F (7, 325) = 2.360, *p* = 0.000	F (7, 853) = 7.365, *p* = 0.000

** *p* < 0.01, —indicates that there was no linear regression between the two variables.

**Table 3 behavsci-13-00736-t003:** Modulating effect analysis result.

Variable	PCS			MCS		
	Model 1	Model 2	Model 3	Model 4	Model 5	Model 6
Public sports space perception	0.103 **	0.059 **	0.058 **	0.151 **	0.152 **	0.153 **
Digital integration	—	0.186 **	0.186 **	—	0.173 **	0.157 **
Public sports space perception * Digital integration	—	—	0.002	—	—	0.009 *
R2	0.046	0.114	0.115	0.113	0.114	0.118
Adjust R2	0.045	0.112	0.112	0.105	0.104	0.103
F	F (1, 863) = 41.485, *p* = 0.000	F (2, 862) = 55.686, *p* = 0.000	F (3, 861) = 37.191, *p* = 0.000	F (8, 856) = 13.614, *p* = 0.000	F (9, 855) = 12.111, *p* = 0.000	F (10, 854) = 10.917, *p* = 0.000
ΔR	0.046	0.069	0.002	0.113	0	0
ΔF	F (1, 863) = 41.485, *p* = 0.000	F (1, 862) = 66.727, *p* = 0.000	F (1, 861) = 0.292, *p* = 0.589	F (8, 856) = 13.614, *p* = 0.000	F (1, 855) = 0.186, *p* = 0.667	F (1, 854) = 0.268, *p* = 0.605

* *p* < 0.05, ** *p* < 0.01, —indicates that there was no linear regression between the two variables.

**Table 4 behavsci-13-00736-t004:** The results of the mediating role of physical exercise activity in older adults of different ages.

Influence Path	Age	B	95% Boot CI	Inspect the Conclusion	Effect
Public sports space perception→LPA→PCS	40–59	0.003	−0.010~0.028	Not	—
	60–69	0.019	0.002~0.083	Yes	14.12%
	70–80	0.034	0.006~0.140	Yes	22.75%
Public sports space perception→LPA→MCS	40–59	0.012	−0.003~0.037	Not	—
	60–69	0.034	0.002~0.111	Yes	100%
	70–80	0.066	0.023~0.177	Yes	21.68%

— indicates that there was no linear regression between the two variables.

## Data Availability

The data in this study can be provided upon request by sending an e-mail to the corresponding author.

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
