# Peer review of "Association of Public Sports Space Perception with Health-Related Quality of Life in Middle-Aged and Older Adults—Evidence from a Survey in Shandong, China"

_behavsci, 2023, doi:10.3390/bs13090736_

Round 1

Reviewer 1 Report

This is a paper that shows that the positive impression of sports facilities among the elderly encourages exercise and increases happiness. This paper is an empirical study. Empirical studies usually take the style of hypothesis testing. However, no hypothesis is presented in this paper. So the reader does not understand why such an analytical model is adopted. In particular, the first half of the paper does not mention at all why digital integration can be expected to have a mediation effect. As for the comparison by age group, it seems abrupt because it is not mentioned in the first half. You should present your hypothesis with reference to sufficient prior research in the first half, and discuss the interpretation of the results in the discussion. In this way, it is necessary to follow the style of empirical research papers and make significant revisions.

Reviewer 2 Report

I would like to express my gratitude regarding the opportunity to review this manuscript.

The article, at this stage, requires several revisions:

- the title and abstract of the article do not refer to the Chinese population

- Abstract: requires changes, so that references are made to the tools used, but also to some statistical data regarding the group of subjects (average age, other variables invoked)

- to check some wording of some paragraphs, even in the Introduction, because it creates confusion or the sources cited refer to something else.

- the level of knowledge on the topic is not presented. In the Introduction, no references are made to the results of studies on the topic, even on the analyzed Chinese population.

Also here, but especially in the research part, confusion is created and clarifications are needed regarding the analyzed population category: middle-aged and older adults. To clarify what it means in the opinion of the authors middle-aged and older adults. In the literature, this age category is differentiated in a different way than it is done in the article by the authors. If the information in the current version is maintained, the authors will change the title of the article.

- the hypotheses, but also the study variables, are not clearly stated

- in chapter 2.1., the characteristics of the subjects are presented, reference is made to some tables that are not included in the text of the article. There is no information regarding the gender distribution of the subjects. also, the results are not presented in this sense (were differences between genders identified?). Is it a representative sample?

- when presenting the instruments and their factors, they should be detailed with types of activities. In the Discussions chapter, too much is used of "such as walking" or others, but so far no detail has been given that this type of activity is included in the LPA (walking is only a type of activity included). Lines 446, 451, 454, 455 should be reworded (4 times in a paragraph "such as walking" is repeated!)

- by the way, the results, but especially the Discussions, are sent to another age category of the subjects. In addition, many of the studies with which the obtained results are compared refer to a different type than the one of interest - to the elderly, adolescents.

- The Discussions invoke certain aspects that were not previously presented, neither presented on the instruments used and their dimensions, nor statistical data presented. It is about body composition scores - see lines 284-291, 364-367. To have body composition scores, BMI (body mass index) must be calculated. Nowhere in the text until Discussions, in the lines mentioned above, is there any information about this value of this variable, how it was calculated, etc.!

- also in Discussions it is formulated by making references to studies, but no reference is made to it and they are not found in the References list (for example, line 463).

In addition, the authors cite statistical data from an article that does not include those values (see lines 416-417).

- many sources referred to in the text, including in the comparison of the results in the Discussions, refer to the elderly (18, 20, 21) or teenagers (20).

- in line 449, a possible hypothesis of the differences obtained from the perspective of regionality is put forward. To be explained!

- in line 473 is an identical situation, where it is necessary to identify a series of possible explanations for the fact that the results of the study differ from those of other studies.

- the source from 26 did not write the authors' names

- although the text refers to tables and figures, they are not included in the text. Some information seems relevant for the reader to understand. (table S1-S1, figures S1 and S2, tables 4-15/ row 343).

it would be appropriate for the article to be reviewed by an English native

Round 2

Reviewer 1 Report

The authors made appropriate corrections.

Reviewer 2 Report

I appreciate the work of the authors to improve the article after the reviews.

I would have some aspects to note before it can be published:

- the request to modify and introduce more information to the Summary also referred to specifying the 3 instruments used: IPAQ, HRQoL, Public Sports Space Perception Scale.

- the authors' explanations regarding the age category of the subjects is not convincing. older adults refers to over 65 years. To be classified as middle-aged and older adults, they must be up to 65 (40-65 years old). As the average of the group of subjects is 66.84, this means that a larger number of subjects are much older than 65 years. Or, from the group of subjects, those under the age of 65 seem to be very few.

- there are several typing errors (check the lack of spaces between words) - see, for example, lines 71, 83, 98, 123, 505. In line 538, an idiom inside the phrase is capitalized ("this" )

- further, references to tables and figures that cannot be read are kept, some relevant for the reader. See lines 244 – table S1, 278 – figure S2, 285 – Figure S2, 299 – Tables S2-S4.

Maybe these tables include more statistical data (interaction effects of some invoked variables on the dependent ones). If these statistical data are not in these tables, a series of statistical indicators (t-tests, ANOVA, etc.) should be calculated to provide us with this information in order to be able to discuss and compare the results of this study with other studies.
